



# DMS cycling in the Sea Surface Microlayer in the South West Pacific: 2. Processes and Rates

Alexia D. Saint-Macary[1,2], Andrew Marriner[1], Stacy Deppeler[1], Karl Safi[3], Cliff S. Law[1,2]

[1]National Institute of Water and Atmospheric research, Wellington, 6021, New Zealand

[2]Department of Marine Science, University of Otago, Dunedin, 9016, New Zealand

[3]National Institute of Water and Atmospheric Research, Hamilton, 3216, New Zealand

*Correspondence to*: Alexia D. Saint-Macary (alexia.stmac@gmail.com) and Cliff S. Law (cliff.law@niwa.co.nz)

**Abstract.** As the sea surface microlayer (SML) is the uppermost oceanic layer and differs in biogeochemical composition to

the underlying subsurface water (SSW), it is important to determine whether processes in the SML modulate gas exchange, particularly for climate reactive gases. Enrichment of dimethyl sulfide (DMS) and its precursor dimethylsulfoniopropionate (DMSP) have been reported in the SML, but it remains unclear how this is maintained whilst DMS is lost to the atmosphere. To examine this, a comprehensive study of DMS source and sink processes, including production, consumption and net response to irradiance, were carried out in deck-board incubations of SML water at five locations in different water masses in

the South West Pacific east of New Zealand. Net consumption of DMSP and production of DMS in the light and dark occurred at all sites. The net response of DMS and DMSP to irradiance varied between stations but was always lower than conversion of DMSP to DMS in the dark. In addition, DMS photolytic turnover was slower than reported elsewhere, which was unexpected given high light exposure in the SML incubations. Although no relationships were apparent between DMS process rates and biogeochemical variables, including chlorophyll-*a*, bacteria and phytoplankton group, net bacterial DMSP consumption was

correlated with DMSP and DMS concentrations, and also dinoflagellate and *Gymnodinium* spp. biomass, supporting the findings of a companion study that dinoflagellates play an important role in DMS cycling in the SML. However, net DMS production rates and accumulation were low relative to calculated air-sea DMS loss, confirming that the DMS cycling within the SML is unlikely to influence regional DMS emissions.

## 1    Introduction

The climate reactive trace gas dimethyl sulfide (DMS) is the primary natural aerosol precursor (Yu and Luo, 2010; Leaitch et al., 2013; Park et al., 2017; Sanchez et al., 2018) that contributes to the regulation of climate via formation of cloud condensation nuclei (Charlson et al., 1987; Quinn and Bates, 2011). DMS concentration in the surface mixed layer and emission to the atmosphere are the net result of production and consumption by a variety of biological, photochemical and physical processes (Stefels et al., 2007). DMS is mainly produced by enzymatic cleavage of its precursor

dimethylsulfoniopropionate (DMSP), with acrylate as the other product (Keller et al., 1989; Stefels et al., 2007). DMSP





concentration in seawater is determined by phytoplankton biomass and speciation (Keller et al., 1989) and also bacterial composition and production (Curson et al., 2017), and occurs in dissolved and particulate forms with the latter accounting for ~80% of total DMSP (Keller and Korjeff-Bellows, 1996; Belviso et al., 2000; Yang et al., 2005a; Zhang et al., 2009). It is the dissolved DMSP that constitutes the source for bacterial conversion to DMS, but phytoplankton also release DMS directly

during senescence-related cell-lysis (Yoch, 2002; Stefels, 2000). There are at least four independent pathways by which DMSP can be degraded enzymatically by bacteria, three of which lead to the production of DMS; yet, DMS production represents only 5 to 10% of the available DMSP, as the primary DMSP removal pathway of bacterial demethylation results in production of methanethiol (MeSH) (Kiene and Linn, 2000). As DMS production is influenced by phytoplankton, its concentration in the euphotic zone generally reflects the vertical distribution of primary production and biomass, with a DMS maximum in near-

surface waters and concentration decreasing with depth (Dacey et al., 1998; Bouillon et al., 2002; Rellinger et al., 2009). The main DMS sink in the surface mixed layer is biological consumption, which accounts for 50 to 88% (Galí et al., 2013), with photochemical oxidation and emission to the atmosphere accounting for 8 – 34% and 4 – 6% of DMS loss, respectively (Galí and Simó, 2015). Both production and loss processes are in turn influenced by environmental drivers, such as irradiance, nutrient concentration, temperature and pH, resulting in regional and temporal variation in DMS concentration (Stefels et al.,

45  2007).

The sea surface microlayer (SML) plays a key role in air-sea gas exchange as the interface between the ocean and the atmosphere. It is a very thin layer (1 – 1000 µm) with differing physicochemical and biological properties to the underlaying water (Hunter, 1980), including elevated concentrations of carbohydrates, proteins and lipids (Sieburth, 1983; Cunliffe et al.,

2013). The SML is more biologically active than the underlying subsurface water (SSW), due to high bacterial activity and abundance (Cunliffe et al., 2011). Elevated respiration by bacterioneuston in the SML is reflected in $O_2$ and $CO_2$ emissions (Reinthaler et al., 2008), and altered cycling of trace gases such as $CH_4$, $H_2$, $N_2O$ and CO (Sieburth, 1983; Conrad and Seiler, 1988; Upstill-Goddard et al., 2003). The SML also contributes to climate regulation as a significant source of atmospheric particles and organic aerosol (Leck and Bigg, 2005; Roslan et al., 2010). Dissolved DMSP is often enriched in the SML (Yang

et al., 2005b; Yang and Tsunogai, 2005; Yang et al., 2005a; Zhang et al., 2008; Matrai et al., 2008; Zhang et al., 2009; Yang et al., 2009), potentially due to stabilization of dissolved organic substances (Gibbs adsorption surface, (Adamson and Gast, 1967)) and high surface tension which energetically favours DMSP adsorption (Zhang et al., 2008; Zhang et al., 2009). As a result of elevated dissolved DMSP and enrichment of bacterioneuston in the SML (Sieburth, 1983), DMS production from enzymatic cleavage and also consumption are also elevated in the SML relative to SSW (Yang et al., 2001; Yang et al., 2005b;

Yang et al., 2005a; Yang and Tsunogai, 2005; Zhang et al., 2008). DMS enrichment in the SML is often associated with underlaying phytoplankton blooms dominated by DMSP-producers (Walker et al., 2016) and with high phytoplankton biomass in general (> 2 mg m$^{-3}$ (Yang et al., 2005a; Zhang et al., 2009)). Indeed, DMS enrichment in the SML may require specific biological, biogeochemical and meteorological conditions, which may result in anomalously high air-sea DMS flux in regions





of high productivity (Walker et al., 2016). However, understanding of the factors that maintain DMS enrichment in the SML

is limited, particularly as few studies have examined the biogeochemical composition of the SML.

Drivers of DMSP and DMS cycling are more intense in the SML than the SSW. Wind increases ventilation of DMS from the SML (Yang et al., 2001; Yang and Tsunogai, 2005; Yang et al., 2005a; Zhang et al., 2008) and may also concentrate material in surface patches that act as hotspots for DMS cycling. In addition, incident light and ultra-violet (UV) exposure are greater,

in the absence of water column attenuation (Hardy, 1982; Stolle et al., 2020), which may influence DMSP and DMS both directly and indirectly (Zemmelink et al., 2005; Zemmelink et al., 2006). DMS photo-oxidation to dimethyl sulfoxide (DMSO) under a full light spectrum (Kieber et al., 1996) is enhanced in the presence of photosensitizers, such as chromophoric dissolved organic matter (CDOM) (Brimblecombe and Shooter, 1986; Brugger et al., 1998; Vogt and Liss, 2009), which is generally enriched in the SML (Frew et al., 2002; Frew et al., 2004). Conversely, this may also limit light exposure in the SML as organic

matter enrichment and gel particles may attenuate UV and photosynthetically active radiation (PAR) in the SML (Bailey et al., 1983; Carlucci et al., 1985; Agogué et al., 2005). Irradiance represents a sink for DMS via photo-oxidation, but also stimulates intracellular production of DMSP in phytoplankton under light stress and inhibits the bacterial consumption of DMS (Sunda et al., 2002). Consequently, light may enhance both DMS production and consumption, and so the net effect of these processes may be particularly significant in the SML. Although solar radiation dose is an important factor determining temporal

and spatial variability of DMS in surface water (Simó and Pedrós-Alió, 1999; Simó and Dachs, 2002; Vallina and Simó, 2007; Miles et al., 2009), only two previous studies have considered the impact of light on DMSP and DMS in the SML (Zemmelink et al., 2005; Zemmelink et al., 2006).

*In situ* measurements in the SML and SSW in South West Pacific waters during the *Sea2Cloud* voyage (Sellegri et al., in

revision) identified only minor DMSP and DMS enrichment (see companion paper (Saint-Macary et al., in revision), and will be referred as S-M1 thereafter), in contrast to a previous regional study (Walker et al., 2016), and also measurements in other regions, as synthesised in Walker et al. (2016). The apparent absence of DMS enrichment in the same region, as determined by a new technique for sampling DMS in the SML (S-M1), and the requirement for high DMS production to maintain SML enrichment relative to ventilation losses (Walker et al., 2016) highlights the need for processes studies of DMS in the SML. In

this paper, SML process rates were measured in deck-board incubations of SML water from five stations across different water masses in the South West Pacific east of New Zealand, to determine the controls of DMSP and DMS, and ultimately the significance of DMS cycling in the SML.



## 2    Method

### 2.1    Regional setting

The Sea2Cloud voyage took place on the 16 to 28 March 2020 (austral autumn) around the Chatham Rise, east of New Zealand, onboard R/V *Tangaroa* (Figure 1). The characteristics of the water masses sampled during this voyage and meteorological conditions are summarized in Table 1, and detailed in the Sea2Cloud introduction paper (Sellegri et al., in revision). Six workboat deployments were carried out to sample the SML and SSW in different water mass types: the subtropical front (STF) at stations 1 and 2, subantarctic water (SAW) at stations 3 and 4, and mixed water (Mixed) at station 6 (see Figure 1, Table 1).

The mixed water had elevated nutrient content relative to the subtropical water (STW) reflecting a mixture of coastal and shelf water with STW (Sellegri et al., in revision). A sipper consisting of a silicon tube with multiple inlets (internal diameter 2.2 mm) that floated on the surface with water enabled sampling of 2.4 L of water from the SML, which was drawn up using a peristaltic pump or manually by syringe (S-M1).

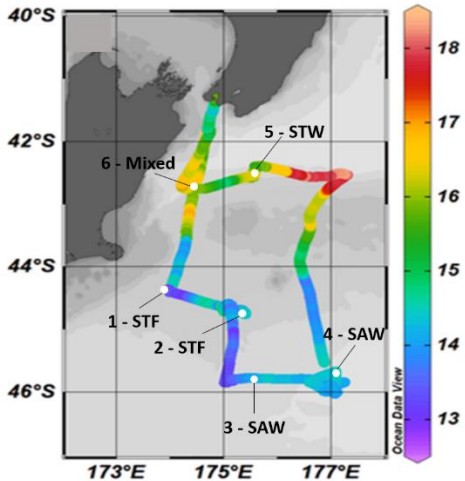

**Figure 1: Sea2Cloud voyage track with workboat station positions overlain on sea surface water temperature (ºC). Figure plotted using Ocean Data View, Schlitzer and Reiner (2020). The grey shading shows the undersea topography, with the darker grey band along 43.5°S indicating the Chatham Rise. Station 5-STW was sampled for SML characterisation (S-M1) but had no deck-board incubation.**

**Table 1: Summary of environmental conditions during the workboat deployments. The water side variables were determined using**
**data from the vessel underway system which sampled at 5-m depth, and windspeed was measured by an Automatic Weather Station, 25.2 m above the sea level.**

| Date | Workboat station and water masses | Workboat sampling time $t_0$-$t_{end}$ | Average wind speed ($\pm$sd) previous 12 h (m s$^{-1}$) | PAR ($\mu$M m$^{-2}$ s$^{-1}$) | Temperature (°C) | Salinity | Chl-*a* at 5 m ($\mu$g L$^{-1}$) | Dominant phytoplankton group (carbon biomass) at 5 m |
|---|---|---|---|---|---|---|---|---|
| 18 March | 1-STF | 0900-1050 | 3.79 ($\pm$2.20) | 481 $\pm$ 576 | 13.03 | 34.55 | 1.54 | Diatom |
| 19 March | 2-STF | 0830-1034 | 7.50 ($\pm$0.87) | 101 $\pm$ 32 | 14.15 | 34.44 | 3.64 | Diatom |
| 21 Mar | 3-SAW | 1020-1159 | 7.88 ($\pm$2.54) | 315 $\pm$ 263 | 13.37 | 34.33 | 0.37 | Dinoflagellate |
| 23 March | 4-SAW | 0845-1022 | 7.36 ($\pm$2.56) | 185 $\pm$ 154 | 13.94 | 34.36 | 0.43 | Dinoflagellate |
| 26 March | 6-Mixed | 0950-1138 | 8.19 ($\pm$3.55) | 582 $\pm$ 478 | 16.24 | 34.78 | 0.89 | Diatom |



## 2.2    Deck incubation set up

Each deck incubation was carried out after workboat SML sampling, as described in S-M1, except at 5-STW where no incubation was carried out as sampling occurred in the afternoon in contrast to the other stations. The SML water was

transferred by gravity into pre-rinsed and flushed 6 x 250-mL UV transparent borosilicate glass bulbs that transmit 90% of UV-A and UV-B, with the bulbs filled completely to eliminate any headspace. The bulbs were incubated in a shallow 37-L seawater bath (17-cm depth), half-immersed in continually flowing surface water to maintain temperature whilst maximizing irradiance to mimic the SML. PAR light Odyssey® photosynthetic irradiance recording system were placed next to the deck incubation to record incoming irradiance in the wavelength range 400 – 700 nm.


Each of the five deck incubations was of 6-hour duration (from midday or 1400 to 1800 or 2000) and used DMS process rate measurement techniques (Simó et al., 2000; Yang et al., 2005b; Yang and Tsunogai, 2005; Yang et al., 2001), modified to small water volumes. Three treatments were each carried out in duplicate. The first pair of bulbs (A) were exposed to ambient deck irradiance to simulate *in situ* conditions in the SML but excluded air-sea loss. The second treatment (B) was maintained

in the dark with the bulbs covered by black tape. Exclusion of light eliminated DMS photo-oxidation and light stress and provided an estimate of the net biological effect on DMSP and DMS in the dark. Light was also excluded in set (C) which included addition of dimethyl disulfide (DMDS), an inhibitor of DMS bacterial consumption, at a final concentration of 200 nmol $L^{-1}$ (Wolfe and Kiene, 1993), so providing a dark DMS production rate.

## 2.3    DMSP and DMS analytical system

Time zero ($T_0$) DMSP and DMS concentrations were determined from the original water sample. After 6 hours incubation, water was sub-sampled from the borosilicate bulbs into 118 mL amber bottles for DMSP and DMS analysis. For DMS measurements, water from the amber bottles was withdrawn in plastic Terumo® syringes and injected through a 25-mm glass microfiber filter (GF/F) into a 1-mL loop, before transfer to a silanized sparging tower where the sample was sparged for 5 minutes with nitrogen ($N_2$) at a flow rate of 50 mL $min^{-1}$. Nafion® dryers removed water vapor from the gas samples before

DMS preconcentration at −110 ℃ on a Tenax® trap. The trap was then heated to 120 ℃ to release the DMS onto an Agilent Technology 6850 Gas Chromatography coupled to an Agilent 355 Sulfur Chemiluminescent Detector (GC-SCD). The daily sensitivity and detection limit of the detector were confirmed using VICI® methyl ethyl sulfide and DMS permeation tubes, with an average detection limit of 0.14 (± 0.03) pgS $sec^{-1}$. For DMSP measurements, 20-mL glass vials were filled and 2 pellets of NaOH added before gas-tight sealing the vials. DMSP samples were stored in the dark at ambient temperature with analysis

within 24 hours of sampling, using the semi-automated purge and trap system and GC-SCD as described above. A wet standard calibration curve was made daily from a stock solution of DMSP diluted in Milli-Q®, with calibration concentrations ranging from 0.1 to 95 nmol $L^{-1}$. These were decanted into 20-mL gas tight glass vials, hydrolysed with 2 pellets of NaOH, and then injected into the sparging unit and processed as samples.



## 2.4 Rate calculation

The rate of change in DMSP and DMS concentration, $k$ , was calculated from the linear slope between $T_0$ and $T_6$ hours and converted to per day rates (nmol $L^{-1}$ $d^{-1}$), with turnover time (d) subsequently calculated by dividing the initial DMS/P concentration by the rate, as described in Table 2. The incubation design had some limitations, with only 2 data points and a short incubation time; however, the 6-hour period was compatible with natural light availability and minimised bottle effects. The net irradiance response of DMSP, $k_{DMSP\ ir}$, and DMS, $k_{DMS\ ir}$, were calculated as the differences between the set exposed

to light (A) and dark (B). The net DMSP dark bacterial consumption rate, $k_{DMSP\ cn}$, was calculated using the change in DMSP concentration in set B over the 6-hour incubation. Net DMSP dark bacterial consumption rate has been previously calculated using a first order loss rate constant as the slope of the natural log of DMSP concentration versus time (Kiene, 1996); however, as there were only 2 time points ($T_0$ and $T_6$) the slope of the linear decrease in DMSP concentration was used in the current study. The net DMS dark bacterial consumption rate, $k_{DMS\ cn}$, was calculated as the difference between the dark sets (C) and

(B), with and without DMDS addition, respectively (Yang et al., 2005b; Yang et al., 2001). The DMS dark production rate, $k_{DMS\ pr}$, was estimated as the change in DMS concentration in the DMDS treated samples in set (C) (Yang et al., 2005b; Yang et al., 2001; Simó et al., 2000). DMS dark yield was calculated as the ratio between the DMS dark production rate and the DMSP dark bacterial consumption rate. Process rates were compared with the calculated DMS air-sea flux (S-M1) and a DMS air-sea turnover, $\tau_{a/s}$, was also generated by relation to the initial DMS concentration in the SML.


**Table 2: Definition and calculation of DMSP and DMS process rates and turnovers.**

| Process | Abbreviation | Process calculation (nmol $L^{-1}$ $d^{-1}$) | Turnover (d) |
|---|---|---|---|
| DMSP dark bacterial consumption rate | $k_{DMSP\ cn}$ | (DMSP slope set B) | $\tau_{DMSP\ cn}$ |
| Net irradiance response rate of DMSP | $k_{DMSP\ ir}$ | (DMSP slope set A – DMSP slope set B) | $\tau_{DMSP\ ir}$ |
| Net irradiance response rate of DMS | $k_{DMS\ ir}$ | (DMS slope set A – DMS slope set B) | $\tau_{DMS\ ir}$ |
| DMS dark production rate | $k_{DMS\ pr}$ | (DMS slope set C) | $\tau_{DMS\ pr}$ |
| DMS dark yield | DMS dark yield | ($k_{DMS\ pr}$ / $k_{DMSP\ cn}$) | |
| DMS dark bacterial consumption rate | $k_{DMS\ cn}$ | (DMS slope set C – DMS slope set B) | $\tau_{DMS\ cn}$ |
| DMS air-sea flux | $F_{SML}$ | See S-M1 | $\tau_{a/s}$ |

## 2.5 Statistical analysis

The Shapiro test was used to verify the normality of variable distribution. For the non-normally distributed variables Spearman's rank correlation was carried out, and for the normally distributed data a Pearson test was applied. Linear correlation

was considered significant where the coefficient of correlation (rho for Spearman's rank and r for Pearson test) was higher than 0.5 and p-value was lower than 0.05.



# 3    Results

## 3.1    DMSP process rates

DMSP concentrations decreased over the 6 hour incubation in all treatments, with highest losses at the frontal stations and
lowest at 3-SAW and 6-Mixed (Suppl. Info. Figure S1). The DMSP loss was generally similar for all treatments within each
station, except for 3-SAW which showed higher DMSP loss in set B. Although variable between stations, $k_{DMSP\ ir}$ was negative
at 3 stations (range: $-13$ to $+29$ nmol L$^{-1}$ d$^{-1}$ average: $+7$ nmol L$^{-1}$ d$^{-1}$), with the lowest rate at 4-SAW and highest at 2-STF
(Figure 2a). The DMSP dark bacterial consumption rate was generally greater than $k_{DMSP\ ir}$, with an average loss of
53 nmol L$^{-1}$ d$^{-1}$ (range: 13 – 97 nmol L$^{-1}$ d$^{-1}$), with the lowest at 6-Mixed and highest rates in STF waters. As a result, there
was a net loss of DMSP at all stations (average 47 nmol L$^{-1}$ d$^{-1}$; range: 9 – 101 nmol L$^{-1}$ d$^{-1}$). The DMSP data from the deck
incubation are summarized in Table 3, with the rates also considered in terms of turnover of DMSP concentration in the SML
(T$_0$ concentration), as described in the Methods. DMSP dark bacterial consumption turnover ($\tau_{DMSP\ cn}$) was faster than $\tau_{DMSP\ ir}$
with average values of 1.1 d (range: 0.7 – 1.4 d) and 7.3 d (1.7 – 16 d), respectively (Figure 2b). $\tau_{DMSP\ cn}$ was fastest in STF
water but relatively uniform across the other stations, whereas $\tau_{DMSP\ ir}$ did not show any pattern in relation to water mass.
Overall, only $k_{DMSP\ cn}$ showed to be correlated to biogeochemical variables in the SML, such as dinoflagellate and
*Gymnodinium* biomasses, DMSP and DMS concentrations, and the >50 µm phytoplankton size fraction (Suppl. Info. Table
S1).

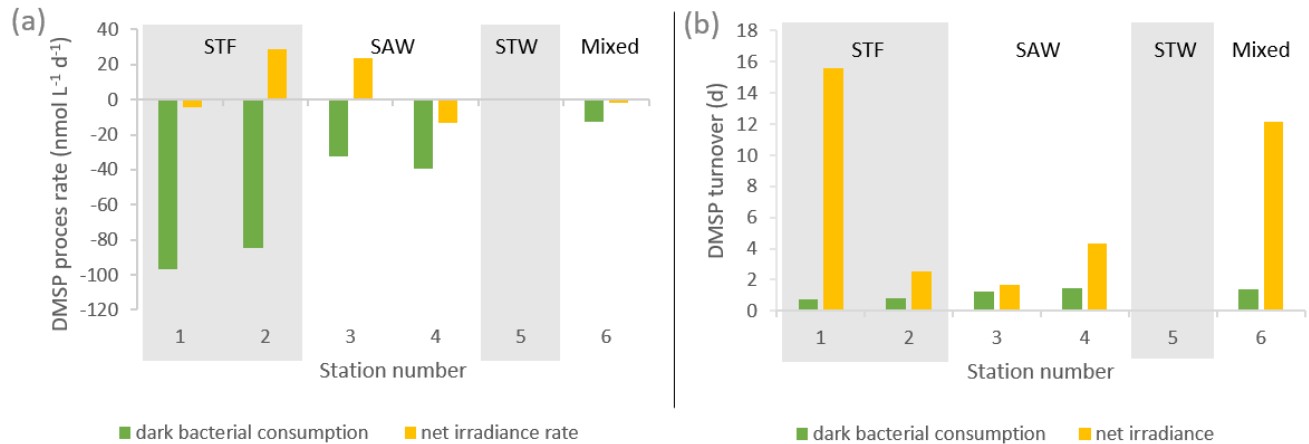

**Figure 2: (a) DMSP process rates (nmol L$^{-1}$ d$^{-1}$), and (b) DMSP consumption and net irradiance turnover (days) determined in deck**
**incubation of SML water at each station. Water mass type is indicated by the label at the top of the figure and also the shading.**
**Station 5-STW was sampled for SML characterisation (S-M1) but had no deck-board incubation.**

## 3.2    DMS process rates

In contrast to DMSP, DMS concentration increased in all incubations with significant differences between stations (Suppl.
Info. Figure S2). Station 2-STF showed the largest increase in DMS in set A relative to T$_0$ (8 nmol L$^{-1}$ d$^{-1}$), whereas there were



only minor increases at 1-STF and 6-Mixed (< 1 nmol L$^{-1}$ d$^{-1}$). There were also variations within stations, with DMS increases in the dark treatment (set B) at 2-STF, 3-SAW and 4-SAW (Suppl. Info. Figure S2). Dark production was the dominant DMS process at an average of 3 nmol L$^{-1}$ d$^{-1}$, exceeding $k_{\text{DMS cn}}$ and $k_{\text{DMS ir}}$ at all stations except 6-Mixed (Figure 3a). The $k_{\text{DMSP cn}}$ varied from 0 to 4.44 nmol L$^{-1}$ d$^{-1}$ and was higher at 1-STF and 4-SAW. $k_{\text{DMS ir}}$ was positive and also highest at 2-STF, whereas it was negative at 3-SAW and 4-SAW. DMS dark yield was on average 6%, with maximum yield at 4-SAW (16%) and

minimum at 6-Mixed (1.4%, Table 4).

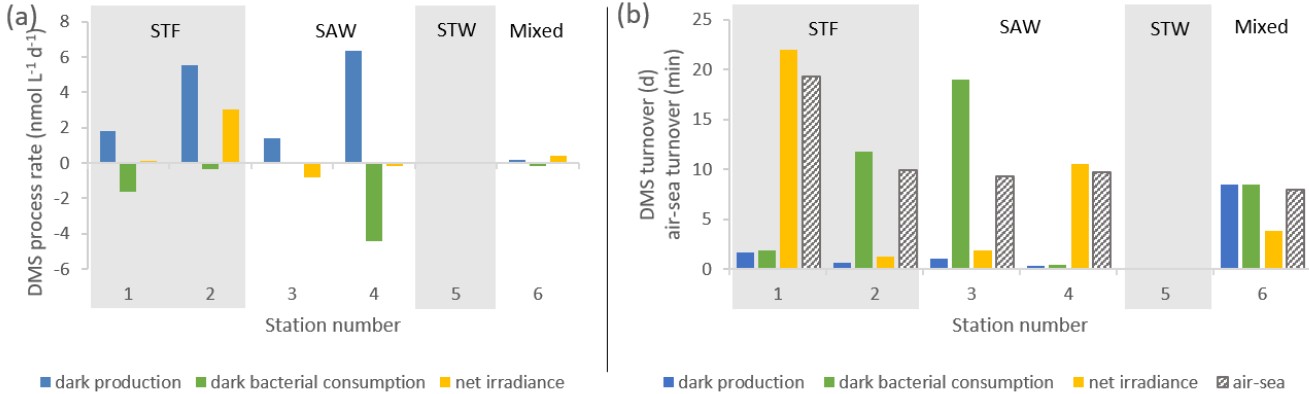

**Figure 3: (a) DMS process rates (nmol L$^{-1}$ d$^{-1}$) at each sampling station. (b) DMS turnover time in days for dark production, dark bacterial consumption and net irradiance response, and in minutes for air-sea turnover, with water mass type indicated by the label at the top of the figure and also the shading. Station 5-STW was sampled for SML characterisation (S-M1) but had no deck-board**

**incubation.**

The DMS rates were assessed in relation to DMS concentration to generate a turnover, as described in the Methods section and summarized in Table 4. $\tau_{\text{DMS cn}}$ varied between 0.4 and 19 d with an average of 8.3 d (Figure 3b) and was generally similar to $\tau_{\text{DMS ir}}$ (average 7.9 d; range 1.2 – 22 d). DMS dark production turnover was faster than $\tau_{\text{DMS ir}}$ and $\tau_{\text{DMS cn}}$ at all stations, except 6-Mixed, at an average of 3 d (range: 0.3 – 8.4 d). However, the air-sea turnover $\tau_{\text{air-sea}}$, calculated from the air-sea flux

(S-M1), was considerably shorter at an average 11 min (8 – 19 min), and so ~ 1,100 fold faster than $\tau_{\text{DMS pr}}$. In addition, DMS process rates and turnover did not show any significant correlations with ancillary variables (chl-$a$, phytoplankton community composition, bacterial abundance (S-M1, correlation coefficients in Suppl. Info. Table S1).

**Table 3: Summary of DMSP process rates (nmol L$^{-1}$ d$^{-1}$) and turnover (d) in SML water for each station. The DMSP concentration**

**is in nmol L$^{-1}$. The calculation details for the rates and turnovers are given in Table 2.**

| Date | Station # | [DMSP]$_{\text{SML}}$ | $k_{\text{DMSP cn}}$ | $k_{\text{DMSP ir}}$ | $\tau_{\text{DMSP ir}}$ | $\tau_{\text{DMSP cn}}$ |
|------|-----------|-----------------------|----------------------|----------------------|-------------------------|-------------------------|
| Mar-18 | 1-STF | 69.75 | −97 | −4.5 | 16 | 0.7 |
| Mar-19 | 2-STF | 72.13 | −84 | 29 | 2.5 | 0.9 |
| Mar-21 | 3-SAW | 40.06 | −33 | 24 | 1.7 | 1.2 |
| Mar-23 | 4-SAW | 56.38 | −40 | −13 | 4.3 | 1.4 |
| Mar-26 | 6-Mixed | 17.95 | −13 | −1.5 | 12 | 1.4 |
| *Average* | - | *51.25* | *−53* | *6.7* | *7.3* | *1.1* |



**Table 4: Summary of DMS process rates (nmol L$^{-1}$ d$^{-1}$), turnover (d) and air-sea turnover (min) in SML water for each station. The DMS concentration is in nmol L$^{-1}$. The calculation details for the rates and turnovers are given in Table 2.**

| Date | Station # | [DMS]$_{SML}$ | $k_{DMS\ pr}$ | $k_{DMS\ cn}$ | $k_{DMS\ ir}$ | Net accumulation rate | DMS dark yield | $\tau_{DMS\ cn}$ | $\tau_{DMS\ pr}$ | $\tau_{DMS\ ir}$ | $\tau_{a/s}$ |
|---|---|---|---|---|---|---|---|---|---|---|---|
| Mar-18 | 1-STF | 3.08 | 1.8 | −1.6 | 0.1 | 0.3 | 1.9 | 1.9 | 1.7 | 22 | 19 |
| Mar-19 | 2-STF | 3.76 | 5.6 | −0.3 | 3.1 | 8.3 | 6.6 | 12 | 0.7 | 1.2 | 10 |
| Mar-21 | 3-SAW | 1.52 | 1.4 | −0.1 | −0.8 | 0.7 | 4.3 | 19 | 1.1 | 1.9 | 9.3 |
| Mar-23 | 4-SAW | 1.69 | 6.4 | −4.4 | −0.2 | 1.8 | 16.1 | 0.4 | 0.3 | 10.6 | 9.7 |
| Mar-26 | 6-Mixed | 1.52 | 0.2 | −0.2 | 0.4 | 0.4 | 1.4 | 8.4 | 8.4 | 3.8 | 8.0 |
| *Average* | - | *2.31* | *3.1* | *−1.3* | *0.5* | *2.3* | *6.1* | *8.3* | *3.1* | *7.9* | *11* |

## 3.3 DMS:DMSP ratio

The DMS:DMSP ratio in sets A (light) and B (dark) were compared to the initial *in situ* DMS:DMSP in the SML (incubation T$_0$, Figure 4). DMS:DMSP was on average 0.05 in the SML (range: 0.03 – 0.08), and increased to 0.07 in the absence of air-sea loss in the incubations in both set A and B (ranges 0.04 – 0.09 and 0.05 – 0.08, respectively). DMS:DMSP was similar in set A and B at each station, except for 3-SAW where it was higher in set B (dark).

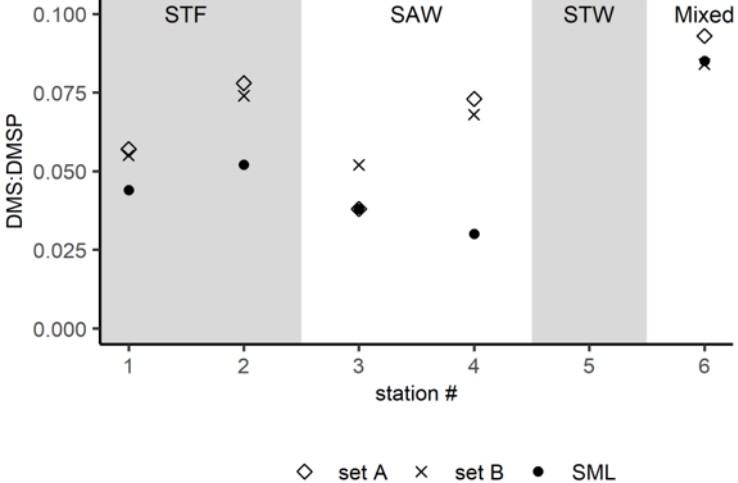

**Figure 4: DMS:DMSP ratio in the SML at T$_0$, and at the end of the 6 hour deck incubations in set A and B at each station. At 3-SAW, the ratio in set A is equal to, and so obscured, by the initial ratio in the SML. Station 5-STW was sampled for SML characterisation (S-M1) but had no deck-board incubation.**

## 4 Discussion

The characteristics of the SML during the current study contrasted with a previous regional study (Walker et al., 2016), and
results from other regions (Nguyen et al., 1978; Yang, 1999; Yang and Tsunogai, 2005; Yang et al., 2005a; Zhang et al., 2009;





Zemmelink et al., 2006), with only limited enrichment of DMSP, DMS and chl-*a* at one of six stations (S-M1). Consequently, the DMS air-sea flux was not significantly affected by DMS in the SML (S-M1), and was generally consistent with the climatological estimates of Lana et al. (2011) and Wang et al. (2020). Although DMS dark production was the dominant process in the deck-board incubations, net DMS accumulation was low (Table 4) which, combined with the reported variation

in enrichment, raises questions as to how excess DMS is maintained in the SML when air-sea loss is significant (Figure 3b). The following discussion considers the processes and factors influencing DMSP and DMS cycling in the SML, and whether these are sufficient to generate DMS enrichment (S-M1).

**Table 5: Summary of DMS and DMSP processes in the SML and potential factors influencing EF DMS in different water masses. In the EF columns, depletion is indicated by "−", enrichment by "+", and "n/s" is not significant. Station 3-SAW where DMS**
**enrichment occurred is highlighted in the shaded row. DMSP enrichment was measured at station 5-STW, and so is not presented in this Table. For the dominant phytoplankton group in the SML, "D" and "F" stands for diatom and dinoflagellate, respectively. For the processes, a "−" indicates DMS loss and a "+" indicates DMS production, with triplicate symbols indicating the dominant DMS/P transformation process at the respective station (air-sea flux was always a loss process for DMS, and always exceeded other process rates). The maximum air-sea flux (>5 μmol m$^{-2}$ d$^{-1}$) is indicated by 2 "−" signs. Results from SM-1 are indicated by * and**
**n/d stands for "not determined".**

| Station number | EF DMSP* | EF DMS* | Dominant phyto-plankton* | DMS: DMSP | DMS dark yield (%) | DMS air-sea flux* | $k$ DMS pr | $k$ DMS cn | $k$ DMS ir | $k$ DMSP cn | $k$ DMSP ir |
|---|---|---|---|---|---|---|---|---|---|---|---|
| 1-STF | − | − | F | 0.04 | 1.9 | − | +++ | − − | + | − − | − |
| 2-STF | − | − | n/d | 0.05 | 6.6 | − − | +++ | − | ++ | − − | + |
| 3-SAW | − | + | F | 0.04 | 4.4 | − | +++ | − | − − | − − | + |
| 4-SAW | − | − | F | 0.03 | 16 | − | +++ | − − | − | − − | − |
| 6-Mixed | − | n/s | F | 0.08 | 1.4 | − | + | − | ++ | − − | − |

## 4.1 DMSP processes

The current study is, to our knowledge, the first to determine DMSP process rates in the SML. DMSP loss occurred in all treatments at all stations, with the highest $k$ DMSP cn in the diatom bloom at 2-STF. As chl-*a* and bacterial abundance were elevated at this station (S-M1), DMSP loss was enhanced in association with elevated biological activity; however, bacterial
community composition is considered a more significant determinant of $k$ DMSP cn than bacterial abundance (Vila-Costa et al., 2008). In the current study, $k$ DMSP cn was correlated to the dinoflagellate and Gymnodinium biomass in the SML, and to DMS/P concentrations in the SML, confirming the importance of dinoflagellate on DMS/P dynamics in the SML (Suppl. Info. Table S1, S-M1).The $k$ DMSP cn, range of 13 – 97 nmol L$^{-1}$ d$^{-1}$ measured in the SML is higher than regional rates determined in SSW with the $^{35}$S-DMSP method (3 – 60 nmol L$^{-1}$ d$^{-1}$; Lizotte et al. (2017)), consistent with the SML being more biologically active
than the SSW (Cunliffe et al., 2011). However, this difference in regional consumption rates may reflect methodological differences, as the net concentration change method used in the current study generally delivers higher consumption rates than the dissolved $^{35}$S-DMSP method (Vila-Costa et al., 2008). That dark bacterial consumption was the dominant DMSP process is consistent with bacterial demethylation being the primary DMSP removal process in the surface ocean (Kiene and Linn, 2000).




The net response of DMSP to irradiance was variable (Figure 2a and Table 5), as reported in other studies (Slezak et al., 2001; Slezak et al., 2007). Exposure to light can affect intracellular synthesis of DMSP (Stefels, 2000; Hefu and Kirst, 1997), and phytoplankton DMSP production is enhanced during antioxidant response to light stress (Sunda et al., 2002). Exposure to UV and UV + PAR may have differential effects on intracellular accumulation of DMSP in the coccolithophore *Emiliania huxleyi*
(Sunda et al., 2002; Van Rijssel and Buma, 2002; Archer et al., 2010), with DMSP synthesis inhibited under high UV radiation (Archer et al., 2018; Herndl et al., 1993; Muller-Niklas et al., 1995; Slezak et al., 2001; Sunda et al., 2002; Van Rijssel and Buma, 2002). However, UV radiation may also inhibit bacterial DMSP removal (Slezak et al., 2001), and result in DMSP accumulation. As the response of phytoplankton DMSP synthesis and bacterial cycling varies with light intensity, and light exposure also varied between deck incubations, this limits interpretation of the irradiance-related processes and factors
influencing DMSP cycling in the SML. Regardless, the net effect of irradiance on DMSP was minor relative to dark bacterial consumption, indicating potential for DMS production in the SML.

### 4.2 DMS production and bacterial consumption

DMS dark production was the dominant DMS process in the SML, whereas the net response to irradiance and dark bacterial consumption of DMS varied between stations, with no single factor responsible for this variation (Figure 3, Table 4.1). DMS
is produced by enzymatic cleavage of DMSP and also direct phytoplankton release (Yoch, 2002), as supported by the correlations between DMSP and DMS concentration in both the SSW and SML (S-M1). The mean $\tau_{DMS\ pr}$ and $\tau_{DMS\ cn}$ in the SML were 3 d and 8.3 d, within the range reported elsewhere for the SML (0.1 to 4.2 d; (Yang et al., 2005b; Yang and Tsunogai, 2005; Yang et al., 2005a; Zhang et al., 2008)). In terms of DMS consumption, $\tau_{DMS\ cn}$ in the SML was more rapid than $\tau_{DMS\ cn}$ reported for subsurface water during SOAP (2.3 to 36.5 d; (Lizotte et al., 2017), again reflecting faster biological
turnover in the SML (Yang and Tsunogai, 2005; Yang et al., 2005a; Zhang et al., 2008).

### 4.3 Effect of irradiance on DMS

As with DMSP, the response to irradiance was variable, with a net decrease in DMS at 3-SAW and 4-SAW in set A (Figure 3, Table 5), suggesting photo-oxidation of DMS (Brimblecombe and Shooter, 1986), and a positive effect at the other stations indicating stimulation of DMS production. This is consistent with the higher DMS:DMSP ratio in the light incubation (set A)
at most stations, indicating elevated DMS production or inhibition of DMS bacterial consumption by light (Figure 4). That the net response of DMS to irradiance was negative only at the SAW stations suggests differing sensitivity to light between water masses, although no significant relationships were identified between PAR and DMS concentration, process rates or enrichment in the SML (S-M1, Suppl. Info. Table S1). Under light stress, phytoplankton may elevate DMS production via three pathways - overflow, antioxidant system and cell damage (Gali et al., 2013). Under stress, such as nitrogen-limited
conditions with the overflow hypothesis (Stefels, 2000), and iron limitation with the antioxidant pathway (Sunda et al., 2002), excess intracellular DMSP is produced and released, so increasing substrate for conversion to DMS. In surface water exposed to high solar radiation with low UV attenuation the cell damage pathway may result in increased cell permeability, further





increasing DMSP availability for conversion to DMS (Gali et al., 2013). An additional impact of irradiance is the inhibition of bacterial DMS consumption (Slezak et al., 2007; Toole et al., 2006a), which may enhance DMS accumulation (Gali et al.,
2013). As the net response of DMS to irradiance was more often positive (Figure 3, Table 5) this indicates that biological responses, such as stress production of DMS and inhibition of DMS consumption had greater losses to photo-oxidation.

With the exclusion of air-sea exchange in the deck-board incubation, DMS:DMSP in set A and B would be expected to exceed the *in situ* ratio in the SML, as observed at most stations (Figure 4). Only 3-SAW showed a different trend, with a similar
DMS:DMSP in set A to the *in situ* ratio but higher ratio in set B, potentially indicating suppression of DMS production by irradiance at this station. Determination of the DMS photolysis constant, which is the inverse of $\tau_{DMS\ ir}$, generated rates of $0.004 - 0.035$ $h^{-1}$, which are significantly lower than rates reported for subsurface water ($0.026$ to $0.14$ $h^{-1}$ (Toole et al., 2006b; Brimblecombe and Shooter, 1986; Brugger et al., 1998; Kieber et al., 1996). This slower photolytic DMS turnover was unexpected due to the elevated solar and UV radiation exposure in the SML, although this may reflect variability of irradiance
both during and between the deck incubation, in contrast to laboratory studies that use constant radiation often with wavelength cut-offs (Brimblecombe and Shooter, 1986; Brugger et al., 1998; Kieber et al., 1996; Toole et al., 2006a). In addition, previous photolysis studies have used filtered seawater (Kieber et al., 1996; Toole et al., 2006a; Brimblecombe and Shooter, 1986; Brugger et al., 1998), in contrast to the unfiltered samples in this study in which particle scattering and absorption may have buffered photolytic DMS losses. The slower photolytic DMS turnover from the current study can also be due to the SML
biogeochemical properties; often the SML is enriched in gel-like particles which can protect the SML compounds from high solar irradiance (Ortega-Retuerta et al., 2009).

## 4.4  DMS dark yield

Notwithstanding differences between the $^{35}$S-DMSP and dark net loss methodologies (Vila-Costa et al., 2008) DMS dark yields in the SML were in agreement with previous regional estimates (Lizotte et al., 2017; Vila-Costa et al., 2008). DMS dark yield
was highest at 4-SAW due to high $k_{DMS\ pr}$ (Table 5), potentially due to the elevated dinoflagellate and small flagellate biomass at this station (S-M1), although no relationships were identified between DMS dark yield and other variables (see Suppl. Info. Table S1). The DMS:DMSP ratio indicated that 5 to 10% of DMSP was converted to DMS, consistent with previously reported estimates, and supporting the hypothesis that the proportion of DMSP cleaved to DMS is relatively constant across the ocean and independent of regional influences and phytoplankton composition (Lizotte et al. (2017) and references therein). Although
this is surprising considering the reported enrichment of bacteria and dissolved DMSP in the SML (Yang et al., 2009; Yang et al., 2005b; Zhang et al., 2008; Matrai et al., 2008), it is consistent with the general absence of enrichment in the current study (S-M1, see Table 5).




### 4.5 Relating SML processes to DMS enrichment

Air-sea emission was the dominant process controlling DMS concentration in the SML, with the air-sea turnover rate in the
SML, calculated from the air-sea flux in S-M1, ranging from 8 to 19 min which is within the range reported in other studies
(0.1 – 24.4 min (Yang et al., 2005a; Yang and Tsunogai, 2005; Yang et al., 2001)). Consequently, despite net DMS
accumulation in the SML (Table 4), the significantly greater air-sea loss should deplete DMS in the SML and so prevent
enrichment (Table 5). As the SML DMS production rates in the current study are consistent with others reported (Yang et al.,
2005b; Yang and Tsunogai, 2005; Yang et al., 2001; Zhang et al., 2008) DMS production in the SML is unlikely to match air-
sea loss, and consequently, regional DMS air-sea flux should not be influenced by DMS cycling in the SML (Yang and
Tsunogai, 2005). Furthermore, at the single station where DMS enrichment was significant (3-SAW, S-M1), DMS production
did not dominate and the DMS dark yield was low (Table 5), and so the enrichment cannot be explained. It should be borne in
mind that the apparent disconnect between SML process rates and enrichment may reflect comparison of *in situ* conditions
with artificial conditions in the deck-board incubation. It is challenging to simulate the SML *in vitro*, particularly in recreating
the SML thickness and interaction with the overlaying atmosphere and subsurface water, and the incubation design may have
introduced artefacts (particle concentration, wall effects) and altered light exposure and attenuation relative to *in situ*
conditions.

The current study was motivated by previous regional observations of high DMS enrichment and the associated influence of
the SML on air-sea DMS emissions (Walker et al., 2016), but has found no evidence to support this. The previous study noted
the large inconsistency between measured DMS production rates in the SML and inferred production rates required to support
air-sea flux estimates. This inconsistency is further confirmed by net DMS accumulation rates of 0.3 – 8 nmol L d$^{-1}$ in the
current study, that are consistent with previous regional estimates (mean 15 nmol L d$^{-1}$, (Lizotte et al., 2017)). The significant
correlation between $k_{DMSP\,cn}$ with DMS and DMSP concentration, dinoflagellates and *Gymnodinium* biomass (Suppl. Info.
Table S1, S-M1), confirms that this phytoplankton group and species play are important determinants of DMSP and DMS
cycling in the SML. This further emphasises the requirement for an optimal combination of biogeochemical, physical and
meteorological factors - low winds, near-surface stratification and a bloom of high-DMSP dinoflagellates - for significant
DMS enrichment to occur in the SML (S-M1), as during the SOAP voyage (Walker et al., 2016). The combined observations
from S-M1 and the current study confirm that SML DMS enrichment is rare in the South-west Pacific, reflecting that DMSP
and DMS cycling in the SML are insufficient to maintain DMS enrichment concurrent with elevated air-sea loss.

**Acknowledgment.** We would like to thank Theresa Barthelmeß for her contribution to the SML sampling, and Antonia Cristi
and Wayne Dillon for their help during the Sea2Cloud campaign. This research was supported by NIWA SSIF funding to the
Ocean-Climate Interactions Programme. We would also like to thank the support and expertise of the Officers and Crew of
the R/V Tangaroa.
**Author contribution.** Alexia D Saint-Macary developed the experiment set up. Alexia Saint-Macary wrote the manuscript,
analysed DMSP and DMS. Andrew Marriner contributed to DMSP and DMS analysis. Stacy Deppeler analysed samples on



the Flowcam and processed results, and Karl Safi identified the species by optical microscopy. Alexia D Saint-Macary, interpreted the results, with guidance from Cliff Law. There are no conflict of interests.

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
