# Peer review of "DMS cycling in the Sea Surface Microlayer in the South West Pacific"

_EGUsphere, 2022_

## Author Comment (AC1)

Manuscript egusphere 2022-504: **DMS cycling in the Sea Surface Microlayer in the South West Pacific: 2. Processes and Rates**

Answers to Reviewer 1

**General comments regarding the conclusions:**

Even though a number of SML physico-chemical variables were examined in the current study, the lack of replicate information in the form of duplicates six hours apart (T0 and T6) restricts the confidence that can be applied to the findings. Consequently, statements about observed correlations and what they imply need to be made with caution and words such as "confirms" (L 340) & "confirm" (L344) should be avoided. The statement "…the current study confirm that SML DMS enrichment is rare in the SW Pacific" (L344) is based on very limited data made at one location east of NZ in one season that may not be so at other locations at other times of the year in the broad study region.

Answer: The reviewer is correct that there were a limited number of experiments run in just one region of the SW Pacific and so the confidence of the study findings is equally limited. Consequently "confirm" was replaced by "suggest" L348 and L352. This was also confirmed in the additional text in the Conclusion section: L360-361 "Although these results are only representative of one region of the South West Pacific during the austral autumn…"

DMS process rates and turnovers (defined in Table 2) for the three seawater types sampled; however, clear conclusions derived from this analysis are lacking. I would like to see a conclusion (section 5) added that provides concise findings from the complex analysis of data undertaken.

Answer: a conclusion section was added to the manuscript:

"The current study presents the results of a comprehensive investigation into DMSP and DMS processes in the SML that is, to our knowledge, the first to assess DMSP cycling and the net effect of irradiance on both DMSP and DMS in the SML. Bacterial consumption of DMSP and dark production of DMS were the dominant processes in the SML, with irradiance having relatively minor impact on both species. Although these results are only representative of one region of the South West Pacific during the austral autumn, the combination of in situ SML observations in S-M1 and process rates in the current study indicate that DMS enrichment in the SML is rare and that net accumulation of DMS in the SML is insufficient to balance DMS air-sea loss."

This report is identified as a companion to a more substantial part-1 report referred to as S-M1 that is under revision (L85-86). The many references to S-M1 for methodology (e.g. details about the new sipper collection technique, L101-103) and DMS flux results (L158, 205-207) and references to S-M1 throughout the discussion leave the reader wondering about aspects of the study that are not available. The publication status of SM1 is unclear. Is it possible to include an Ocean Science manuscript number or information that it has been or will be accepted for publication at L452?

Answer: the companion paper is currently under review, and can be accessed at https://egusphere.copernicus.org/preprints/2022/egusphere-2022-499/

**Specific comments:**

General answer: the grammatical mistakes and unclear sentences were modified in the manuscript as suggested.

L56-57: I suggest it would be better to say, "due to stabilization by dissolved organic substances …". Please clarify what the dissolved DMSP is adsorbed to. I presume it is the stabilised dissolved organic substances.

Answer: added L58 "high surface tension which energetically favours dissolved DMSP adsorption to dissolved organic substances".

L67: what sort of "material"? particulate organic materials?

Answer: added L70-71 "particulate organic material in surface patches".

L97-99: It is stated here that were six workboat deployments but information is not included for 6 stations in Table 1. If the workboat went out isn't there a sampling time and data from the vessel sampling system for the parameters given in Table 1 for 5-STW? Even though there was no deck-board incubation carried out for station 5 (L107) the reader would expect to see data in Table 1. I suggest that L107 in the caption for Fig 1 and L114-115 are moved together into L97-99 to explain up front the circumstances for station 5 (STW).

Answer: added in Table 1 a line with station 5-STW parameters. A sentence was also added L103-105 to explain why there was no deck-board incubation carried out at station 5-STW.

L135: What type of Tenax? TA? TC?

Answer: "TA" added L140.

L136: Please change "chromatography" to "chromatograph". Was a chromatography column used? If so please specify the column type and the applied chromatographic conditions.

Answer: precision on the chromatograph column added L141 "DB-megabore sulfur SCD column, 70 m length, 0.530 megabore diameter and film thickness 4.30 µm".

L139: It should be explained for those unfamiliar with the analytical approach that DMSP is base catalysed to DMS by the addition of alkali that allows DMSP to be indirectly measured in the form of DMS on a molar conversion basis.

Answer: added L145 "For DMSP measurements, 20-mL glass vials were filled and 2 pellets of NaOH added, to hydrolyse DMSP to DMS".

L141: add "DMS" before "calibration". Add "water" after "Milli-Q". Was it an actual Millipore water system? If not, say "deionised water".
Answer: yes, it was an actual Millipore water system.

L150: There is inconsistency in the description "net DMSP dark bacterial consumption rate," with what is shown in Table 2. L154.
Answer: corrected to "the DMSP dark bacterial consumption rate" L156, and to "The DMS dark bacterial consumption rate" L160.
Table 2: Air-sea turnover in minutes is not shown to be so where turnover is (d).

Answer: added in Table 2 caption "Definition and calculation of DMSP and DMS process rates in nmol $L^{-1}$ $d^{-1}$ and turnovers in d and min for air-sea turnover."

L185: The shading in Fig 2 is confusing because shading is usually used to represent night-time periods on plots. This is most confusing for Fig 4 where light and dark treatments are compared. I recommend that the shading is removed and the water mass types are separated by vertical dashed and/or dotted lines in Figs 2, 3 and 4.
Answer: the shading from figures 2, 3 & 4 were removed and replace by dashed vertical lines. The figure captions were modified accordingly.

Please comment on the significance of the average values shown in Tables 3 &4.

Answer: a sentence was added in the Table 3 and Table 4 captions "Results are the mean value of duplicate incubations."

L233: Please define "EF DMS" here in the caption. Was EF previously defined? I presume it means enrichment factor?

Answer: EF defined L239 "enrichment factor".

L274: Define the "SOAP" experiment and where it was conducted.

Answer: added L279-280 "the Surface Ocean Aerosol Production voyage (SOAP) conducted in the same region of the South West Pacific"

L339: State the correlation value/s.

Answer: added L345 and 347-348 "between $k_{DMSP\,cn}$ with DMS (rho = -0.87; p = 0.05; Spearman's rank correlation; Suppl. Info. Table S1) and $k_{DMSP\,cn}$ with DMSP concentration, dinoflagellates and *Gymnodinium* biomass (r = −0.92; p = 0.03, r = −0.99; p = 0.01, and r = −0.95; p = 0.05, respectively, Pearson tests, Suppl. Info. Table S1, S-M1)".

L346: Please check the spelling of Theresa B?

Answer: the spelling is correct.

Some of the references have DOIs inserted but many do not. Please complete the referencing by including all available DOIs.

Answer: the available DOIs have been added in the references.

The SI requires the same attention to detail as the manuscript. Please include a title and authors heading for the SI document.

Answer: the title of the manuscript, authors' names and affiliations were added.

Fig S1 caption: Please provide a legend to explain each abbreviation and also specify what the different treatments are, and I recommend changing the shading to vertical dashed lines to avoid night/day confusion.

Answer: more explanation in the figure captions were given, such as an explanation of each treatment, definition of the abbreviations. The shaded areas were replaced by vertical dashed lines.

---

## Author Comment (AC2)

Manuscript egusphere 2022-504: **DMS cycling in the Sea Surface Microlayer in the South West Pacific: 2. Processes and Rates**

Answers to Reviewer 2

**Major points**

The incubation experiments are performed under three conditions (a) In presence of irradiance, (b) dark, and (c) dark with DMDS added. The effect of other environmental stressors is not clear. For example, the variation of irradiance between the sites is mentioned as an uncertainty mentioned later in the paper, but should be made clear that it is not considered in the methodology section. It is also not clear what the role of wind is when comparing the onboard experiment with the sea-air fluxes from paper 1. Although the authors mention that the enrichment would not explain the sea-air fluxes of DMS, the contribution factor would also depend on the local wind speed, would it not?

Answer: "However, the variation in PAR between the different experiments was not considered in the interpretation" was added L122-123. Regardless, it is stated in the discussion L287 that "no significant relationships were identified between PAR and DMS concentration, process rates or enrichment in the SML".

The role of wind is not discussed in this paper as the wind was excluded from the deck-board incubations. However, the air-sea DMS flux (determined in SM-1) was much higher, independent of windspeed, than the net DMS accumulation in all experiments in the current study; L326-328 "Consequently, despite net DMS accumulation in the SML (**Error! Reference source not found.**), the significantly greater air-sea loss should deplete DMS in the SML and so prevent enrichment (**Error! Reference source not found.**)". To further emphasise this, the air-sea turnover in Table 4 was expressed in days rather than minutes.

A problem with drawing the main conclusion is the low sample size. Most of the conclusions are derived from just two datapoints per sample (T0 and T6) and hence the validity of the results over a larger number of datapoints, or over different timescales can be questioned. Considering the contrasting results from previous studies in the same study region, one wonders whether those results and the current results are biased by sampling issues. It would be better for the authors to explain more in detail the differences from the earlier studies and the drivers of these differences, rather than say that this study confirms the lack of SML DMS playing a major role in the sea-air flux.

Answer: We acknowledge the limited number of experiments (5) and duplication of each treatment, as discussed above in response to Reviewer 1. The limited SML water volume precluded triplicates and also sampling at more timepoints during the experiments. The inferred DMS production in the SML during SOAP reported in Walker et al (2018) cannot be compared directly as it was estimated using two different approaches (in situ measurements and comparison of calculated and measured DMS flux). However, net DMS accumulation rates from a similar number (6) of deck-board incubations during SOAP were reported in Lizotte et al (2017) and were derived using the same approach of deck-board sampling at T0 and T6 timepoints only. To emphasise this the final paragraph of the Discussion is now rewritten:

"The current study was motivated by previous regional observations of high DMS enrichment and inferred influence of the SML on air-sea DMS emissions during the SOAP voyage (Walker et al., 2016), but has found no evidence to support this. The previous study noted the large inconsistency between estimated SML DMS production rates and the inferred production rates required to support air-sea flux estimates. This inconsistency is further confirmed by the similarity in net DMS accumulation rate of $0.3 - 8$ nmol L $d^{-1}$ (n = 5) between the current study and during SOAP (range -1 to 11 nmol L $d^{-1}$; n=6; Lizotte et al., (2017). The significant correlations observed in the current study between $k_{DMSP\ cn}$ with DMS (r = -0.87; p = 0.05; Spearman's rank correlation; Suppl. Info. Table S1), and also between $k_{DMSP\ cn}$ with DMSP concentration, dinoflagellates and *Gymnodinium* biomass (r = −0.92; p = 0.03, r = −0.99; p = 0.01, and r = −0.95; p = 0.05, respectively, Pearson tests, Suppl. Info. Table S1, S-M1), indicates that phytoplankton community composition is an important determinant of DMSP and DMS cycling in the SML. This then supports the contention that an optimal combination of biogeochemical, physical and meteorological factors - low winds, near-surface stratification and a bloom of high-DMSP dinoflagellates – resulted in the significant DMS enrichment in the SML during the SOAP voyage (Walker et al., 2016), whereas these conditions were not experienced during the current study (SM-1)."

The observations are taken during the austral autumn. Are the results valid for other seasons? The role of light is clearly shown through the incubation experiments and hence there should be a larger effect during the other months, providing that the other parameters stay the same. However, the biogeochemistry of the area could also show a change. Hence the title could be more specific to point the season – for example, 'DMS cycling in the Sea Surface Microlayer during Austral Autumn in the South West Pacific: 2. Processes and Rates'.

Answer: As we have few datapoints for the season and for each water type, it is difficult to expand our results to the season or water types. That the results are only representative of one season is now confirmed in the additional text in the Conclusion section: L360-361 "Although these results are only representative of one region of the South West Pacific during the austral autumn…" and we do not feel that it is necessary to reflect this in the title.

The manuscript is dependent on the publication of the SM1 submitted manuscript. It is not just heavily referenced in this paper, but several details regarding the setup and analysis are missing assuming the publication of the first manuscript. This assumption also means that the manuscript should not be published unless the first one is published.

Answer: See the response above to the similar comment by Reviewer 1. All the necessary information is included in both papers, including the sampling strategy, experimental set up, and the instrumentation set up for DMSP and DMS analysis are described in this paper. The companion paper reviewing process is underway and both will be published together assuming they are accepted.

**Minor points**

Considering this is a sister paper to the first one which is also in review, it would be useful for the reader to highlight in the abstract what the results are with respect to the paper part one and also include what are the main results from the first paper.

Answer: Although they are companion papers both are standalone, with only the comparison of calculated air-sea flux in SM1 with the process rates presented in this paper representing an overlap. Consequently, we prefer not to reference SM-1 in this abstract, and we have instead edited the abstract to say L22 "relative to regional air-sea DMS loss reported elsewhere".

L-28 emission to the atmosphere **is** the net result of production and consumption by **various** biological, photochemical

Answer: modification done.

L59 consumption **'is'** also elevated

Answer: modification done.

L-70,71 **which may, directly and indirectly, influence DMSP and DMS**

Answer: modification done.

L-79 Although solar radiation dose is an important factor **in** determining temporal

Answer: modification done.

L-95 The Sea2Cloud voyage took place **from** the 16 to 28 March 2020 (austral autumn)

Answer: modification done.

Figure 1., color bar title?

Answer: modification done.

L110. Change "5-m" to "5 m". Table 1; Specify time zone.

Answer: modification done.

L-110 **an Automatic Weather Station measured windspeed, 25.2 m above the sea level**

Answer: modification done.

L118 photosynthetic irradiance recording system **was** placed next to the deck

Answer: modification done.

L135 DMS **pre-concentration** at −110 â,, ƒ on a Tenex® trap

Answer: modification done.

L-136, 137 **The detector's daily sensitivity and detection limit were confirmed using VICI®**

Answer: modification done.

L-163,164 Which data is the normally and non-normally distributed data used in this experiment? Please provide proof that it was normally and non-normally distributed considering the small number of samples.

Answer: a table with the results from the Shapiro test was added in Supplementary Information. L167 now "The Shapiro test was used to verify the normality of variable distribution (see results in Suppl. Info. Table SI 1)."

Figures 2a and 2b, also in figures 3a and 3b please mention which set? (A, B, or C).

Answer: in figures 2a-b and figures 3a-b are presented the rate and turnover results. The calculation of these rates are in Table 2, and they do not correspond to a particular set.

Highlight the most important results in tables 3, 4, and 5.

Answer: The results from Table 3 and 4 are already described and presented in the Results section and also in Figures 2 and 3. Table 5 is a summary of the rates that puts them in context of the observed EF, which is also discussed.

Label set A (light) and set B (dark) in figure 4 for quick understanding.

Answer: the label was modified in the figure caption and in the figure legend.

In figure 4, where is set C?

Answer: This has been added to the Figure 4 legend, L228-229 "Set C results are not included as bacterial consumption was inhibited by the addition of DMDS, and do the DMSP:DMS would not be comparable to the unperturbed Set A and B".

L-218 **sets** A and B at each station, except for 3-SAW where it was higher in set B (dark).

Answer: modification done.

L233. Please define "EF DMS."

Answer: modification done.

L-338 current study, **which is** consistent with previous regional estimates

Answer: modification done.

L339. What are the correlation values?

Answer: the correlation values were added L347 and L348-349 "The significant correlation between $k_{DMSP\ cn}$ with DMS (rho = -0.87; p = 0.05; Spearman's rank correlation; Suppl. Info. Table S1) and $k_{DMSP\ cn}$ with DMSP concentration, dinoflagellates and *Gymnodinium* biomass (r = −0.92; p = 0.03, r = −0.99; p = 0.01, and r = −0.95; p = 0.05, respectively, Pearson tests, Suppl. Info. Table S1, S-M1) …".

L-354 There are no **conflicts** of **interest.**

Answer: modification done.

Including all available DOIs in the references – these details are missing in multiple places.

Answer: the available DOIs were added in the references.

---

## Author Response (AR2)

Answers to technical issues of manuscript egu-spere-2022-504

L85-86 and L98-99 and L102 "Selligri et al, in revision" This needs the year of publication instead of "in revision".

Answer: the manuscript is not published yet and is in revision. All the information needed are written in the present manuscript so the reference Selligri et al., was removed.

L86 (Saint-Macary et al., egupshere-2022-499) Similar, needs the year of publication.

Answer: the year of publication was added. However, it is not published yet but available as a preprint.

L109 (in caption Fig. 1) Schlitzer and Reiner (2020) must be R. Schlitzer (2020) Also please correct citation in the reference list

Answer: modification done in the caption and in the reference list.

Table 1 Please delete psu for salinity. Salinity is dimensionless

Answer: psu was deleted.

L301-302 "As the net response of DMS to irradiance was more often positive (Figure 3, Table 5) this indicates that biological responses, such as stress production of DMS and inhibition of DMS consumption had greater losses to photo-oxidation." This sentence is not correct. It says that biological responses had losses to photo-oxidation, which is impossible. Please correct.

Answer: sentence corrected to "As the net response of DMS to irradiance was more often positive (**Error! Reference source not found.**, **Error! Reference source not found.**) this indicates that biological responses, such as stress production of DMS and inhibition of DMS consumption, were greater than losses to photo-oxidation."

L406 Citation should be: Nature Microbiology, 2, 17009, doi:10.1038/nmicrobiol.2017.9, 2017.

Answer: modification done.

L431 and L468 Typo in the title

Answer: modification done.

L482 Please correct citation Schlitzer, Ocean Data View

Answer: modification done.

L483 Any update on in revision?

Answer: the manuscript is still in revision, so it was removed from this manuscript.